# Radial Error Motion Measurement and Its Uncertainty Estimation of Ultra Precision Axes of Rotation with Nanometer Level Precision

**DOI:** 10.3390/mi13122121

**Published:** 2022-11-30

**Authors:** Xingbao Liu, Xiaoting Rui, Liang Mi, Qiang Tang, Heng Chen, Yangqiu Xia

**Affiliations:** 1Institute of Launch Dynamics, Nanjing University of Science and Technology, Nanjing 210094, China; 2National Machine Tool Production Quality Supervision Testing Center (Sichuan), Chengdu 610200, China

**Keywords:** ultra-precision axis of rotation, rotation error, Donaldson reversal method, uncertainty estimation

## Abstract

The radial error is the key performance indicator of ultra-precision axis. In order to measure and evaluate the radial error of ultra-precision axis with nanometer accuracy, a measurement system with an accuracy of nanometer based on capacitive displacement probes and standard spheres is developed. The nonlinearity error of capacitive displacement probes, misalignment error of the probes, eccentric error of standard spheres, error caused by environment temperature change, environment vibration and error separation methods are analyzed and the effects of the above factors are obtained; Multiple measurement examples carried out with the measurement system this paaper constructed indicate the repeatability of the measurement system reaches 10.5 nm and the roundness error of artifact separated is less than 4.03 nm. In order to evaluate the measurement dispersion of the ultra-precision axis radial error, the major uncertainty components and the complete process of the comprehensive evaluation of the measurement uncertainty are proposed. The combined uncertainty of radial error motion measurement of the ultra-precision axis with Donaldson reversal is 31.64 nm (*k* = 2).

## 1. Preface

The ultra-precision rotary axis can realize ultra-precision rotary motion and is the physical benchmark for roundness measurement and angle measurement. It is widely used in roundness meter, coordinate measuring machine, gear measuring machine, laser tracker, bearing measuring machine, etc. it is also the core functional component of ultra-precision machine tools represented by ultra-precision diamond lathe, playing an irreplaceable role in the fields of national defense cutting-edge technology and advanced optical element manufacturing [1,2].

The rotation error of the rotation axis represents the undesired relative displacement of the rotation axis in the sensitive direction during the actual movement, which is the key to the performance of the rotation axis. Accurate measurement of rotation error is a key technology in the design, manufacture, performance evaluation and diagnosis of rotation axis. Traditional axis rotation error is measured by contact or non-contact indicator combined with standard sphere/standard mandrel [3]. In this method, the spindle axis is characterized by an ultra-precision standard sphere and standard mandrel, so as to realize the measurement of rotation error. This method can easily obtain an axis rotation error. However, the fabrication and installation errors of the standard sphere and reference standard make it difficult to achieve nanoscale measurement.

The ultra-precision manufacturing of high—power laser optical element manufacturing, aerospace and other industries, as well as the ultra-precision demand of geometric measurement, the accuracy of the rotary axis has been significantly improved, especially the wide application of gas/liquid hydrostatic bearing, the rotation accuracy of the rotary axis reaches 50 nm or even 15 nm [4,5].The rotation error of the spindle is close to or even less than the installation error and manufacturing error of the benchmark. The traditional measurement method cannot meet the needs of the rotation error detection of the ultra-precision rotary axis. The shape error of the standard sphere can be separated by using the equal error separation technology of inversion method, and the measurement accuracy of rotation error can be improved.

At present, the commonly used measurement methods of rotation error separation of ultra-precision axis of rotation include the inversion method, three-point method, multi-step method and many measurement methods derived on this basis [6,7]. The inversion method can eliminate the error in theory, but because in the process of the measuring sensor and standard sphere, all needs to reverse position, leading to the measurement result contains the standard sphere with sensor reverse position error, so that the measurement accuracy is reduced. Although the multi-step method avoids the shortcomings of the inversion method, its measurement steps are tedious and time-consuming, and it is difficult to apply in practical engineering. The measurement accuracy of the three-point method is better than that of the inversion method, which is inferior to that of the multi-step method, but its practical engineering application difficulty is better than that of the multi-step method [8,9,10,11].

In order to realize the measurement and evaluation of nanometer rotary error of ultra-precision rotary axis, this paper develops a rotation error detection system and method based on capacitive displacement sensor and standard sphere, and analyzes the factors of measuring ambient temperature, environmental vibration, non-linearity of a spherical target of capacitive displacement sensor, eccentricity of standard sphere and sensor misalignment. Based on the measurement system, the measurement repeatability and traceability of the system were analyzed. The repeatability of the measurement system was 25.2 nm, and the roundness error of the separation standard sphere was less than 5.8 nm. In order to evaluate the dispersion of measurement results of ultra-precision rotary axis rotation error, the complete process and main uncertainty components of comprehensive evaluation of measurement uncertainty are proposed. The mathematical model of uncertainty evaluation is constructed, and the expanded measurement uncertainty of 29.5 nm (*k* = 2) is obtained. The evaluation results can be used as the field evaluation of instrument accuracy. This paper analyzes the influence of environmental changes and personnel operation on the measurement results and provides direction guidance for improving the comprehensive measurement accuracy.

## 2. Geometric Significance of Rotation Error and Its Measurement Evaluation

### 2.1. Motion Error of Rotary Axis

The spatial motion error of the rotating axis is the variation of the instantaneous actual rotation axis relative to the average rotation axis. According to the error direction, it can be divided into radial motion error, axial motion error, tilt motion error and angle positioning error in Figure 1. According to the frequency component, it can be divided into the synchronous component and asynchronous component. The synchronous component is the component containing the integer order component in the error motion, and the asynchronous error is the non—integer order component in the error motion [12]. 

### 2.2. Measurement and Evaluation of Rotation Error of Ultra-Precision Rotation Axis

#### 2.2.1. Rotation Error Measurement of Rotation Axis

According to ISO 230-7, two non-contact displacement sensors and standard sphere are used to measure the axis rotation error. The measurement system as shown in Figure 2a is used to collect the error data, and the radial motion radius of the axis is calculated by the following formula:(1)$$r\left(\theta \right)=r_{0}+\Delta x\left(\theta \right)\cos\theta +\Delta y\left(\theta \right)\sin\theta$$

Among them: *θ* is the current angle position of the axis, *r*(*θ*) indicates that the axis is at an angle *θ* radial error at, Δ*x*(*θ*) and Δ*y*(*θ*) are the measured values of displacement sensor.

The least quadratic circle method is used to evaluate the error, and the results of axis motion error are obtained.

#### 2.2.2. Measurement of Rotation Error of Ultra-Precision Rotary Axis by Reversal Method

For the ultra-precision rotary axis, the manufacturing error of the standard sphere is equal to or even greater than the axis rotation error. In order to eliminate the influence of the standard sphere on the measurement results, the inversion method can be used to separate the error. The basic principle of Donaldson inversion method is shown in Figure 3, where the sensor reading before and after inversion is *M*_1_(*θ*) and *M*_2_(*θ*), then the axis rotation error *S*(*θ*) with standard sphere *R*(*θ*) can be obtained by Formulas (2) and (3), and the motion error result [13] can be obtained by correlation error evaluation algorithm.
(2)$$S\left(\theta \right)=\frac{M_{1}\left(\theta \right)+M_{2}\left(\theta \right)}{2}$$
(3)$$R\left(\theta \right)=\frac{M_{1}\left(\theta \right)-M_{2}\left(\theta \right)}{2}$$

## 3. Measurement and Verification of Rotation Error of Ultra-Precision Shafting

### 3.1. Analysis of Influencing Factors of Ultra-Precision Rotary Shafting Test

In order to obtain the measurement ability of better measurement accuracy, this paper analyzes the main factors of the rotation error measurement process of ultra-precision rotary shaft system from the measurement environment, measurement system and other aspects.

#### 3.1.1. Environmental Impact Factors

The environmental factors of the measurement process mainly include environmental temperature and environmental vibration. As a result, the measurement results of the thermal expansion coefficient are different from that of the environment. When the measured axis is still, the author continuously monitors the change of displacement between the standard sphere and the sensor with the ambient temperature. The results are shown in Figure 4a. Within 20 m of the test, the thermal drift of the structure reaches 0.35 μm when the ambient temperature fluctuates by 0.5 °C. The maximum thermal drift of the structure is 62 nm at any one minute.

The influence of environmental vibration mainly includes foundation vibration and motor running vibration. The displacement change between standard sphere and sensor is continuously monitored and frequency spectrum analysis is carried out, as shown in Figure 4b. The limit value of vibration displacement affected by environmental vibration is less than 10nm, which mainly exists in three frequency bands of 0~5 Hz, 20~25 Hz and 45~65 Hz.

The environmental factor test shows that the environmental temperature and vibration should be strictly controlled in the test of axis rotation accuracy. At the same time, the axis should be driven to run stably for more than 30 m. The measurement system should be fully isothermal treatment to reduce the influence of environmental factors.

#### 3.1.2. Nonlinear Error of Displacement Sensor Measurement

In the ultra-precision axis of rotation error measurement, the non-contact and high-resolution capacitive displacement sensor are used to measure the rotation error to avoid the influence of the contact displacement sensor measurement force on the measurement result. As shown in Figure 5, the capacitive displacement sensor is designed according to the principle of idealized plate type pole distance change. When working, the probe is used as an electrode, and the measured conductive object is used as a relative electrode. The movement of the measuring object causes the change of plate distance, thus causing the change of capacitance value [14,15].

In the measurement of axis of rotation, the spherical target is generally used. In this case, the measuring surface is spherical, and the electric field line is bent, which makes the output characteristics of the sensor vary with the radius of curvature. The influence law of the output characteristics of the sensor is analyzed by experiment and finite element simulation [16].

(a)Nonlinear error measurement of spherical target by displacement sensor

In order to study the nonlinear characteristics of the sensor moving relative to the spherical target in the measurement, the experiment as shown in Figure 6 is designed. In the experiment, the spherical target is moved in X direction and Y direction, respectively. The displacement sensor is used to measure the moving distance of the spherical target and the reading of the sensor is observed. 

The curve shown in Figure 7 shows that the nonlinear error of sensor gains corresponding to 1 μm is 0.41 nm when the Φ0.5 mm probe and the standard sphere Φ25 mm are used and the initial clearance is 20 μm, and the axial nonlinearity can be neglected in the measurement of the rotation error of nano-level ultra-precision axis of rotation.

(b)Nonlinear error of lateral displacement of spherical target measured by sensor

It can be seen from Figure 5 that the output characteristics of the sensor will change due to the change in the sensing area and the deflection of the electric field line. When the sensor is offset, as shown in Figure 8, its output characteristics will also change. Using the test device shown in Figure 6, test the sensor output when moving laterally at different offset distances, as shown in Figure 9a. The horizontal axis in Figure 9a represents the lateral offset distance of the displacement sensor shown in Figure 8. The vertical axis in Figure 9a identifies the reading of the displacement sensor shown in Figure 8.

The nonlinear characteristics of the sensor can be obtained by polynomial fitting: (4)$$y=0.00083{\left(x-0.90\right)}^{2}+0.289$$

Without losing generality, it can be written as follows:(5)$$y=ax^{2}+c$$
where *a* and *c* are the determination coefficients related to the characteristics of the sensor.

The test results show that the sensor output has a significant nonlinear with the offset distance, and the influence of the sensor lateral offset should be paid attention to when measuring the ultra—precision rotating axis accuracy.

In view of the fact that there is always a certain lateral offset in the actual measurement, the measurement of the displacement sensor relative to the spherical surface under different initial lateral offset positions is tested. The test results are shown in Figure 9b. The horizontal axis of Figure 9b shows the nominal distance of displacement sensing moving along the measurement direction under different initial lateral offset distances. The horizontal axis of Figure 9b shows the measured distance of displacement sensing moving along the measured direction under different initial lateral offset distances. It can be seen from Figure 9b that when the lateral initial offset changes from 0 to 40 μm, the linear gain of the sensor remains unchanged. Therefore, in order to ensure the linear gain of the measured value to be basically constant, the capacitive displacement sensor should be adjusted to make its lateral offset as small as possible. 

At the same time, due to the existence of lateral offset, the error caused by lateral component should be considered when considering the eccentricity of standard sphere installation and sensor alignment error.

#### 3.1.3. Influence of Eccentric Error of Standard Sphere Installation

The error motion of the axis of rotation has good periodicity and can be written as:(6)$$M\left(\theta \right)=a_{0}+\overset{\infty }{\underset{i=1}{\sum }}\left(a_{i}\cos\theta +b_{i}\sin\theta \right)$$
$$\mathrm{Among}\;\mathrm{them}:\;a_{i}=\frac{2}{N}\sum_{k=1}^{N}m\left(\theta_{k}\right)\cos i\theta_{k}$$
$$b_{i}=\frac{2}{N}\sum_{k=1}^{N}m\left(\theta_{k}\right)\sin i\theta_{k},\;a_{0}=\frac{1}{N}\sum_{k=1}^{N}m\left(\theta_{k}\right)$$
$$m\left(\theta_{k}\right)=m\left(\theta \right)-\alpha_{0}$$

Suppose the eccentricity error is *e*. When the angle deviation is *θ*, the normal component and tangential component of eccentricity error are *e* cos*θ* and *e* sin*θ*. Assuming that the initial tangential offset between the sensor and the standard sphere is $e_{0}$, then the tangential offset between the sensor and the standard sphere is: (7)$$e_{tangential}=e_{0}+e\cos\theta$$

In this case, the displacement sensor error caused by eccentricity can be expressed as follows:(8)$$e_{eccentric\_effect}=e_{tangential\_effect}+e\sin\theta +c$$

Using Formulas (5) and (8), then,
(9)$$e_{eccentric\_effect}=a{\left(e_{0}+e\cos\theta \right)}^{2}+e\sin\theta +c$$

It can be seen from the above equation that the sensor output signal contains two components: first-order error and second-order error. The first-order error can be eliminated by eliminating the fundamental frequency, while the second-order error cannot be eliminated mathematically. If the eccentricity error *e* = 5 μm, the second-order error will reach 10 nm. Therefore, the eccentricity error of the standard ball must be controlled by certain means to minimize it as far as possible in Figure 10.

#### 3.1.4. Influence of Misalignment Error of Displacement Sensor

In the measurement of rotation error, it is assumed that the displacement sensor aims at the center of the standard sphere, but there is misalignment error between the sensor and the standard sphere in the actual measurement that *O_a_* is the center of rotation of the axis to be measured, the radius of the standard sphere is *r*, the misalignment deviation *e*_0_, the motion error vector *E_axis_* of the actual rotation axis, and the rotation speed of the measured axis ω. The displacement sensor reading is: (10)$$m=E_{effect\_mis}+E_{axis\_mis}$$

In Figure 11, among them, *E_axis_mis_* is the component of rotation axis motion error vector on OA axis, *E_axis_mis_* is the tangential offset influence sensor reading component caused by the component of rotation axis motion error perpendicular to OA axis.
(11)$$E_{effect\_mis}=E_{axis}\cos\varphi$$
(12)$$E_{axis_{mis}}=f\left(E_{axis}\sin\varphi \right)$$

Function f(·) denotes the effect of lateral initial offset Equation (5). The inclination angle *φ* is caused by the alignment error. According to the geometric relationship, it can be seen that:(13)$$f\left(E_{axis}\sin\varphi \right)=a{\left(E_{axis}\sin\varphi \right)}^{2}$$
(14)$$m=E_{axis}\cos\varphi +a{\left(E_{axis}\sin\varphi \right)}^{2}$$

Then the error *E_mis_* caused by the misalignment deviation is as follows:(15)$$E_{mis}=m-E_{axis}$$
(16)$$E_{mis}=E_{axis}\cos\varphi +a{\left(E_{axis}\sin\varphi \right)}^{2}-E_{axis}$$

Considering that the radius of the standard sphere (above 25 mm) is much larger than the misalignment deviation *e_0_* (micron scale), the inclination angle *φ* is small. When the tilt Angle φ is small enough, we can get:(17)$$E_{mis}=a{\left(E_{axis}\varphi \right)}^{2}$$

Axial error motion, radial error motion and standard sphere eccentric motion will cause mis center deviation, resulting in sensor output signal error. When the axial error motion value is 0.4 μm and the transverse initial offset is 20 μm, the maximum misalignment error will reach 13 nm, which will affect the measurement of ultra-precision spindle rotation error. 

#### 3.1.5. Influence of Positioning Deviation of Inversion Method

It is assumed that when the inversion method is used to measure the rotation accuracy, the angular deviation between the actual position and the ideal position of the standard sphere after the inversion is φ, where *R(θ)* is the roundness of the standard sphere. As shown in Figure 12a below, the measurement error of rotation accuracy caused by the positioning deviation φ is:(18)$$E\left(\theta \right)=\frac{R\left(\theta \right)-R\left(\theta +\varphi \right)}{2}$$
(19)$$E\left(\theta \right)\approx -\frac{\varphi }{2}R^{\prime }\left(\theta \right)$$

At the same time, the probe is required to rotate 180° relative to the spindle rotor when using the reversal method, and the probe angle position error will be introduced into the measurement signal, and the measurement schematic diagram is shown in Figure 12b. The measurement error is as follows:(20)$$M_{2}\left(\theta \right)=R\left(\theta -\varphi \right)-S_{x}\left(\theta \right)cos\varphi +S_{y}\left(\theta \right)sin\varphi$$
(21)$$E\left(\theta \right)=\frac{M_{1}\left(\theta \right)-M_{2}\left(\theta \right)}{2}-S_{x}\left(\theta \right)$$
(22)$$E\left(\theta \right)=\frac{1}{2}\left[R\left(\theta \right)-R\left(\theta -\varphi \right)+S_{x}\left(cos\varphi -1\right)-S_{y}sin\varphi \right]$$
(23)$$E\left(\theta \right)\approx \frac{1}{2}\left[\varphi \left(R^{\prime }\left(\theta \right)-S_{y}\right)-\frac{1}{2}S_{x}\left(\theta \right)\varphi^{2}+ο\left(\varphi \right)+ο\left(\varphi^{4}\right)\right]$$
where $S_{x}$ and $S_{y}$ is the radial error of X direction and Y direction. When the angle φ is relatively small, the measurement error can be simplified as follows: (24)$$E\left(\theta \right)\approx \frac{1}{2}\left[\varphi \left(R^{\prime }\left(\theta \right)-S_{y}\right)\right]$$

### 3.2. Rotation Error System and Measurement Test of Ultra-Precision Rotary Shaft System

#### 3.2.1. Rotation Error System of Ultra-Precision Rotary Shaft System

Based on the above analysis, an error measurement system for the ultra-precision rotary axis is constructed in Figure 13. The system is composed of centering components, encoder, capacitive displacement sensor and standard sphere. Based on the error processing algorithm of the inversion method, the error analysis software system is developed by using C# framework, which is used to evaluate the full attitude error of the rotary axis and draw the results.

During the test, firstly, the axis of rotation was uniformly rotated at low speed, and the eccentricity of the standard sphere with the axis of rotation was adjusted to be less than 1 μm. Secondly, adjust the displacement sensor to the standard sphere high point; Then, after the axis under test is run at a constant speed for 30 m to reach thermal stability, the encoder index signal is used as the sampling start signal, and the encoder pulse is the sampling external clock. Twenty turns of data were measured each time to obtain *M*_1_(*θ*). After the acquisition is completed, the standard sphere and sensor are rotated 180°, and the acquired data *M*_2_(*θ*) is measured again. *S*(*θ*) and *R*(*θ*) can be obtained by using the correlation error processing algorithm. 

#### 3.2.2. Measurement Repeatability

For a measurement system, its measurement repeatability is an important embodiment of measurement precision, so the author uses the developed measurement system to carry out the repeatability test. Since the rotation error of the axis and the roundness error of the standard sphere are obtained by the inversion method, the key to the repeatability analysis of the detection system is to analyze the repeatability of the detection results of the radial motion error of the axis and the roundness error of the standard sphere. 

Therefore, under the condition of repeatable test, standard sphere A is used to test the rotation accuracy of the same axis. The roundness error curve of the standard sphere and the radial motion error curve of the axis obtained from five tests are shown in Figure 14. Meanwhile, the evaluation results of the roundness error of the standard sphere and the radial motion error of the axis are shown in Table 1.

In conclusion, when the same standard sphere is used to measure the same sample, the standard deviation of roundness error of standard sphere is 4.03 nm, and the standard deviation of radial motion error of axis is 10.5 nm.

#### 3.2.3. Measurement traceability

The traceability of the measurement system is the basis of the system value transmission. The radial motion error and standard sphericity error of the axis can be obtained by using the reversal method to measure the motion error of the ultra—precision rotary shaft system. Considering the stability of the standard sphere and its better measurement technology, we use the standard sphere to transfer and trace the value of the measurement system.

In order not to lose generality, the following two experiments are designed to explore the measurement traceability of the system. ① The roundness error of the standard sphere is compared with the calibration value when the same standard sphere is used for different rotation errors of the measured object. ② Comparison of radial motion error results of the same sample rotation accuracy measured by different standard spheres. 

①
**
*The same standard sphere test for different detection objects*
**


The standard sphere of the detection system is calibrated by a third-party calibration laboratory, and its roundness is 70 nm. Then, an ultra-precision turntable, an aerostatic spindle 1, an aerostatic spindle 2 and an aerostatic spindle 3 are used as the test objects. The axial radial motion error and standard spherical roundness are tested by the reverse method. In order to analyze the deviation between the standard sphere and the calibration value of different measuring objects, the test site and results are shown in Figure 15 and Figure 16 and Table 2.

According to Figure 16, when the same standard sphere is used to test the radial motion error of different samples, the roundness shape of the separated standard sphere is the curve with three peak characteristics, and the maximum deviation of the roundness of the standard sphere obtained is 17 nm compared with the calibration value of the third-party calibration laboratory. 

②
**
*Test of different standard spheres for the same test object*
**


In order to analyze the deviation of the radial motion error of the separated samples when using different standard spheres to test the rotation accuracy of the same measuring object, the deviation is analyzed. Taking an ultra-precision turntable as the test object, different standard spheres (including standard sphere A′s factory roundness error is 50nm, standard sphere B′s calibration roundness error is 70 nm), the test results are shown in Figure 17 and Table 3.

According to Table 3, the maximum difference between the radial motion errors of the axis separated by standard sphere a and standard sphere B is 30.3 nm, and the difference of the average radial motion error of the axis obtained by using the two standard spheres is 4 nm. According to Figure 17, when standard sphere A and standard sphere B are used to detect the rotation error of the same sample, the shape of the radial motion error curve of the separated samples is similar. 

## 4. Evaluation of Uncertainty in Measurement of Rotation Error of Ultra-Precision Rotary Shafting

Measurement uncertainty is a parameter that represents the dispersion of the measured value reasonably given and is the most important basis to measure the credibility of measurement results [17,18]. Therefore, this paper evaluates the measurement uncertainty of ultra-precision rotary error measurement. 

### 4.1. Source Identification and Main Components of Measurement Uncertainty

According to the analysis of the influence of the main measurement factors in the measurement process, the following sources of measurement uncertainty are considered in this paper:The measurement uncertainty introduced by measurement environment mainly includes environmental vibration, environmental temperature change, and environmental electromagnetic interference factors.The measurement uncertainty introduced by the measuring instrument includes the non-linear error of the sensor measuring sphere, the measurement error of the sensor, the inclination error of the sensor installation, etc.Uncertainty of measurement method: data processing related factors, such as eliminating eccentricity, error separation algorithm, etc.

According to the analysis and test in Section 2.1. of this paper and the properties of various uncertainty components, various uncertainty components are evaluated.

#### 4.1.1. Measurement Uncertainty Component Introduced by Measurement Environment

(1)The uncertainty is introduced by the temperature component of the measurement

In the process of measurement, the relative displacement of the measured axis and the measuring system is caused by factors such as the ambient temperature and the heating of the equipment motor, which will affect the measurement results. According to the analysis in this paper, in a complete measurement, the maximum drift of the measurement value caused by the ambient temperature fluctuation is 62 nm. If the uniform distribution is processed, the measurement uncertainty caused by the environmental temperature change is *uEthermal* = 62/2$\sqrt{3}$ = 17.9 nm.

(2)Uncertainty component introduced by environmental vibration

In the process of measurement, the relative displacement of the measured axis and measurement system is caused by the vibration of the environment, which affects the measurement results. According to the analysis in Section 2.2.1 of this paper, it can be seen from the field measurement that the limit value of vibration displacement caused by environmental vibration is less than 10 nm, then it can be considered that the error caused by vibration obeys the arcsine distribution in the interval with half width of 10 nm, and the measurement uncertainty component caused by the vibration is *u_Evibration_* = 10/$\sqrt{3}\;$= 5.78 nm.

#### 4.1.2. Measurement Uncertainty Components Caused by Measuring Instruments

(1)Uncertainty component introduced by sensor′s own measurement error

The maximum indication error of the capacitive displacement sensor after calibration is 20 nm, and the measurement error of the sensor itself obeys the uniform distribution, so the uncertainty caused by the sensor error can be set to follow the uniform distribution with half width of 20 nm, and the calculation result of its component is *u_probe_* = 20/$\sqrt{3}\;$= 11.55 nm.

(2)Uncertainty component introduced by standard sphere

The roundness of the standard sphere after calibration is 70 nm, and the measurement error of the sensor itself obeys the uniform distribution, then the uncertainty caused by the sensor error can be set to obey the uniform distribution with half width interval of 70 nm, and the calculation result of its component is *u_artifact_* = 70/$\sqrt{3\;}$= 40.41 nm.

(3)Uncertainty components caused by nonlinearity of sensor spherical target measurement

It is assumed that the nonlinear measurement error of the spherical target measured by the sensor with nonlinear measurement accuracy *α_0_* is uniformly distributed in the half-width *α*_0_/2 interval. It is known from the experiment that when the initial gap is 20 μm, the nonlinear error of sensor gain corresponding to 1μm movement is 0.41 nm. In the actual measurement, the moving distance between the sensor and the spherical target is less than 1μm, so the uncertainty component of the nonlinear error of the spherical target measured by the sensor is *u_Plateral_* = *α*_0_/2/$\sqrt{3}\;$= 0.12 nm.

(4)Uncertainty component caused by sensor installation tilt error

When the radial tilt error of sensor installation is less than ±0.5°, the maximum axial error is less than 0.4 μm, and the transverse initial gap is 20 μm, the influence of sensor installation eccentricity on output is less than 20 nm, and the sensor installation error is uniformly distributed, so the measurement uncertainty component caused by it is *u_Ptile_* = 20/$\sqrt{3\;}$= 5.78 nm.

(5)The component of uncertainty caused by sensor reversal positioning deviation

The inversion positioning of the sensor is determined by the symmetrical distribution of the relative sphere. The inversion positioning deviation can reach 0.05 mm, and the beating of the standard sphere is less than 1.5 μm in the measurement process, so the measurement deviation is 0.94 nm. Then, the measurement deviation caused by the sensor inversion positioning deviation should follow a uniform distribution with a half-width of 0.94 nm, and its measurement uncertainty component is *u_Rprobe_* = 0.94/$\sqrt{3}$ = 0.54 nm.

(6)The uncertainty component caused by the standard sphere reversal positioning error

The reversal of the standard sphere through tooling reverse benchmark plate, the rotary tooling, such as the maximum deviation of 1° inversion Angle positioning, according to the standard sphere roundness identification result set standard surface for 3 cycle of sine wave, is set under the premise of its biggest roundness is 100 nm has led to the maximum error of standard sphere reversal locating errors of 0.7 nm. Assuming that the measurement error of the inverted positioning deviation from the standard sphere follows a uniform distribution of 1.4nm in full width, the uncertainty component caused by the error is *u_Rartifact_* = 1.4/$2\sqrt{3}\;$= 0.41 nm.

#### 4.1.3. Measurement Uncertainty Components due to Repeatability

Under the condition of repeated measurement, the range of the measurement results is 27.6 nm (Table 1). According to the evaluation method of class A measurement uncertainty, the range coefficient is 2.33, then the measurement uncertainty caused by repeatability is as follows: *u_Repeatability_* = 27.6/2.33 = 11.85 nm.

### 4.2. Evaluation of Measurement Uncertainty

(1)Evaluation of measurement uncertainty of original data

According to the source analysis of measurement uncertainty, the measurement model of the original data can be obtained as follows:(25)$$M\left(\theta_{k}\right)=Probe\left(\theta_{k}\right)+\delta_{artifact}+\delta_{Plateral}+\delta_{Ptilt}+\delta_{vibration}+\delta_{thermal}$$

Among them, *M* (*θ_k_*) is the sensor reading, *Probe* (*θ_k_*) is the sensor error, *δ_artifact_* is the standard spherical error, *δ_Plateral_* is the spherical target measurement nonlinear error, *δ_Ptilt_* is the alignment error caused by sensor installation, *δ_vibration_* is the environmental vibration error, *δ_thermal_* is the environmental thermal deformation error. It can be seen that the measurement uncertainty of the original data is:(26)$$u_{Md}=\sqrt{u_{probe}^{2}+u_{artifact}^{2}+u_{Plateral}^{2}+u_{Ptilt}^{2}+u_{Evibration}^{2}+u_{Ethermal}^{2}}$$

*u_Md_* = 46.4 nm.

According to the standard, each measurement is averaged at each sample point with 20 measurement turns. Therefore, the standard uncertainty of the original data after averaging is as follows:(27)$$u_{\bar{Md}}=\frac{u_{Md}}{\sqrt{N}}$$

(2)Evaluation of measurement uncertainty for eliminating eccentricity of original data

The average depolarization method is used to eliminate the first harmonic in the original data:(28)$$r\left(\theta_{k}\right)=\bar{M}\left(\theta_{k}\right)-a_{0}-a_{1}cos\theta_{k}-b_{1}sin\theta_{k}+\varepsilon_{0}$$

Among them, $\bar{M}\;\left(\theta_{k}\right)$ is the original measurement data, *a*_1_ and *b*_1_ are the first harmonic coefficients obtained by identification, *ε*_0_ is the debiasing residual caused by the encoder′s segmentation error, *n =* 600 is the number of uniformly sampled points in one week, and *a*_0_ is the DC component of the measurement data. The calculation formula is as follows:(29)$$a_{0}=\frac{1}{n}\sum_{i=1}^{n}\bar{M}\left(\theta_{k}\right)$$

Since each measured point is measured under the condition of independent and equal precision, the measurement points are independent of each other. The standard uncertainty of the results after depolarization is as follows:(30)$$u_{r}=\sqrt{u_{\bar{Md}}^{2}+\frac{1}{N}u_{\bar{Md}}^{2}+\frac{2}{N}u_{\bar{Md}}^{2}+u_{\varepsilon_{0}}^{2}}$$

At the same time, according to the analysis of eccentricity separation error of encoder dividing accuracy, when the encoder repeated dividing error is within 0.1°, and the eccentricity is controlled at 1 μm, *ε*_0_ is about 0.25% of the eccentricity, then *u_ε_*_0_ = (1000 × 0.25%)/2$\sqrt{3}\;$= 0.72 nm.

The measurement uncertainty after depolarization is as follows:(31)$$u_{r}=\sqrt{u_{\bar{Md}}^{2}+\frac{1}{N}u_{\bar{Md}}^{2}+\frac{2}{N}u_{\bar{Md}}^{2}+u_{\varepsilon_{0}}^{2}}$$

(3)Evaluation of measurement uncertainty of rotation error based on inversion

The uncertainty model of error separation of inversion method is as follows:(32)$$S\left(\theta \right)=\frac{r_{1}\left(\theta \right)-r_{2}\left(\theta \right)}{2}+\varepsilon_{ball}+\varepsilon_{sensor}+\varepsilon_{repeatability}$$
where, *r*_1_(*θ*) and *r*_2_(*θ*), respectively, represent the test data of eliminating eccentricity before and after inversion, *ε_ball_* and *ε_sensor,_* respectively, represent the measurement error caused by the positioning deviation between the standard sphere and the probe after inversion, and *ε_repeatability_* represents the measurement error caused by repeatability. Since *r*_1_(*θ*) and *r*_2_(*θ*) affect each other, it is difficult to express the uncertain components related to *r*_1_(*θ*) and *r*_2_(*θ*) by an analytical expression with accurate relative coefficients. Therefore, the following equation is used to approximate them in this paper:(33)$$u\left(\frac{r_{1}\left(\theta \right)-r_{2}\left(\theta \right)}{2}\right)\leq \frac{\left(u_{r}{}_{1}+u_{r}{}_{2}\right)}{2}=u_{r}$$

According to Equation (32), the uncertainty of error separation *u_s_* of inversion method is obtained as follows:(34)$$u_{S}\leq \sqrt{u_{r}^{2}+u_{Rprobe}^{2}+u_{Rartifact}^{2}+u_{Repeatability}^{2}}$$

From the above uncertainty components, it can be seen that if *u_S_* = 15.82 nm, and taking the inclusion factor *k* = 2, the expanded measurement uncertainty of spindle rotation accuracy is shown as follows.
*U_s_* = *ku_S_* = 31.64 nm

The repeatability standard deviation of the measurement system constructed in this paper is 10.05 nm, the standard deviation of the standard ball obtained by separation is 4.03, and the extended uncertainty (*k* = 2) of the rotation accuracy measurement results is 31.64 nm. It provides a method for reliable measurement of the rotation accuracy of ultra-precision spindles of 50 nm and has great application potential.

## 5. Conclusions

(1)In order to realize the measurement and evaluation of nanometer rotary error of ultra—precision rotary axis, this paper develops a rotation error detection system and method based on capacitive displacement sensor and standard sphere, and analyzes the factors of measuring ambient temperature, environmental vibration, non-linearity of spherical target of capacitive displacement sensor, eccentricity of standard sphere and sensor misalignment.(2)Based on the measurement system, the measurement repeatability and traceability of the system were analyzed. The repeatability standard deviation of the measurement system was 10.05 nm, and the standard deviation of the roundness error of the separation standard sphere was 4.03 nm.(3)In order to evaluate the dispersion of measurement results of ultra-precision rotary axis rotation error, the complete process and main uncertainty components of comprehensive evaluation of measurement uncertainty are proposed. The mathematical model of uncertainty evaluation is constructed, and the expanded measurement uncertainty (*k* = 2) is 31.64 nm.

## Figures and Tables

**Figure 1 micromachines-13-02121-f001:**
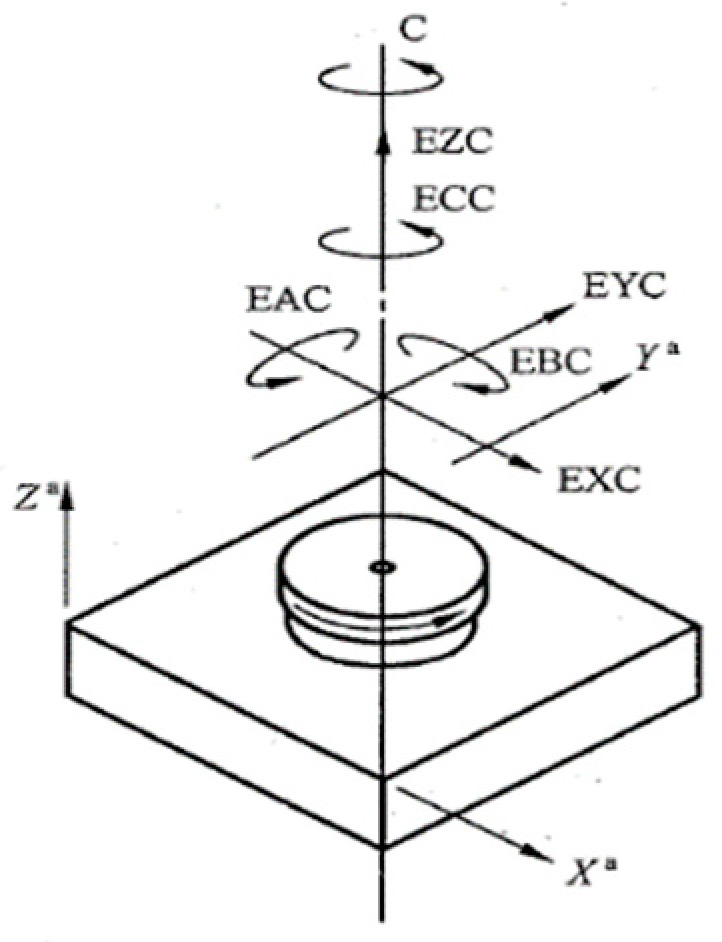
Error motion of rotary axis.

**Figure 2 micromachines-13-02121-f002:**
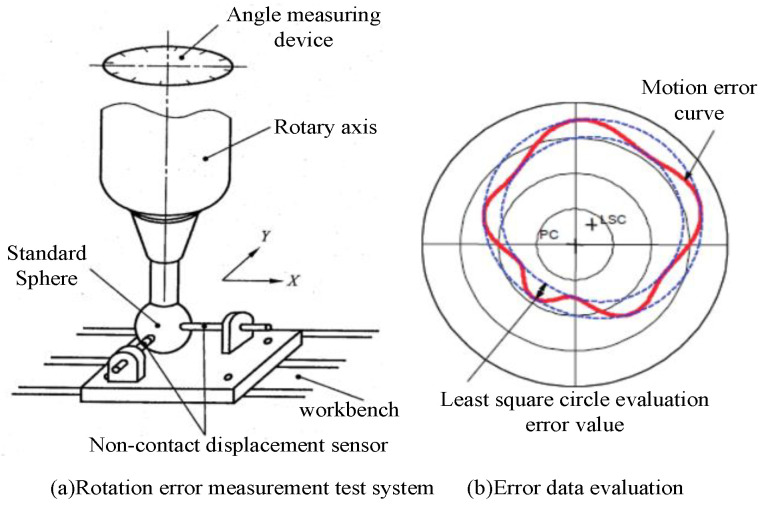
Measurement system and data evaluation of rotary error.

**Figure 3 micromachines-13-02121-f003:**
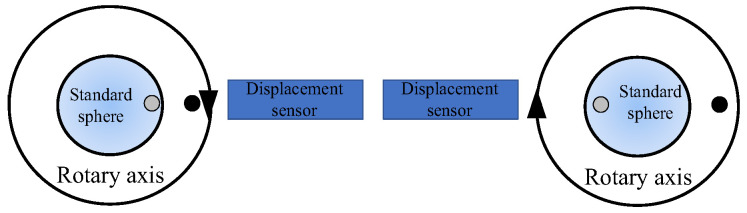
Error separation of inversion method.

**Figure 4 micromachines-13-02121-f004:**
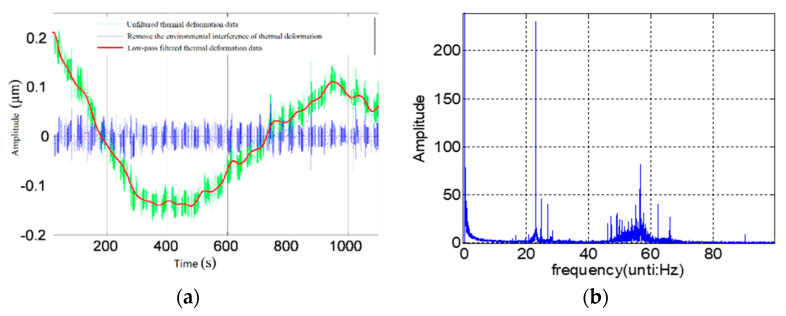
The influence of environmental factors when the spindle (**a**) Influence of ambient temperature, (**b**) Influence of environmental vibration.

**Figure 5 micromachines-13-02121-f005:**
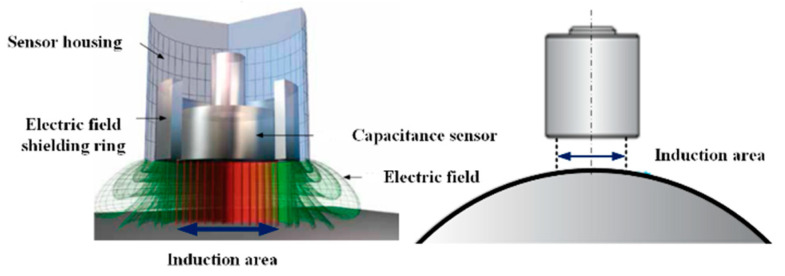
Nonlinear analysis of capacitive displacement sensor.

**Figure 6 micromachines-13-02121-f006:**
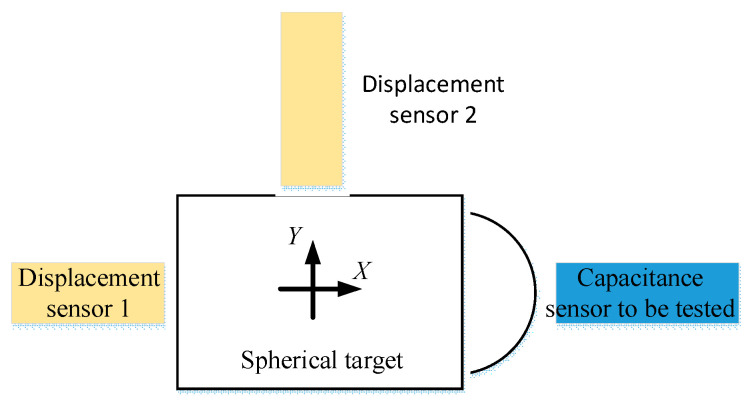
Nonlinear test device of capacitive displacement sensor.

**Figure 7 micromachines-13-02121-f007:**
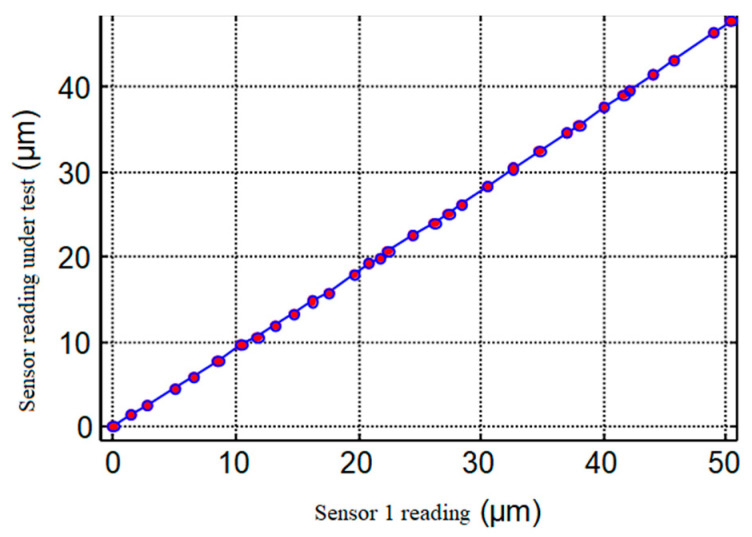
Nonlinear capacitance analysis of displacement sensor.

**Figure 8 micromachines-13-02121-f008:**
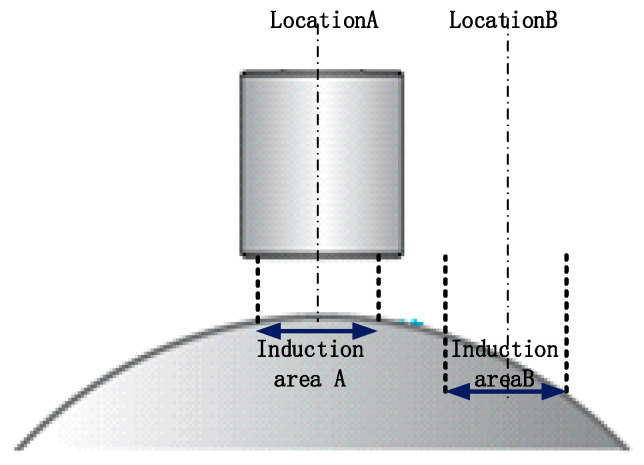
Nonlinear analysis of capacitive displacement sensor.

**Figure 9 micromachines-13-02121-f009:**
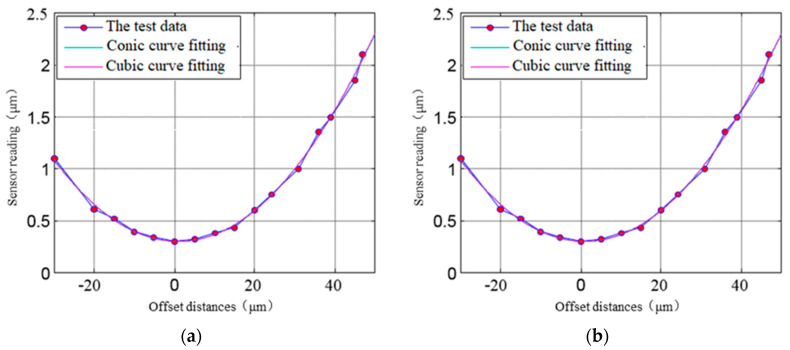
Nonlinear characteristics of lateral offset output of capacitive displacement sensor. (**a**) Sensor output when moving laterally at different offset distances, (**b**) Sensor output under different lateral initial offset.

**Figure 10 micromachines-13-02121-f010:**
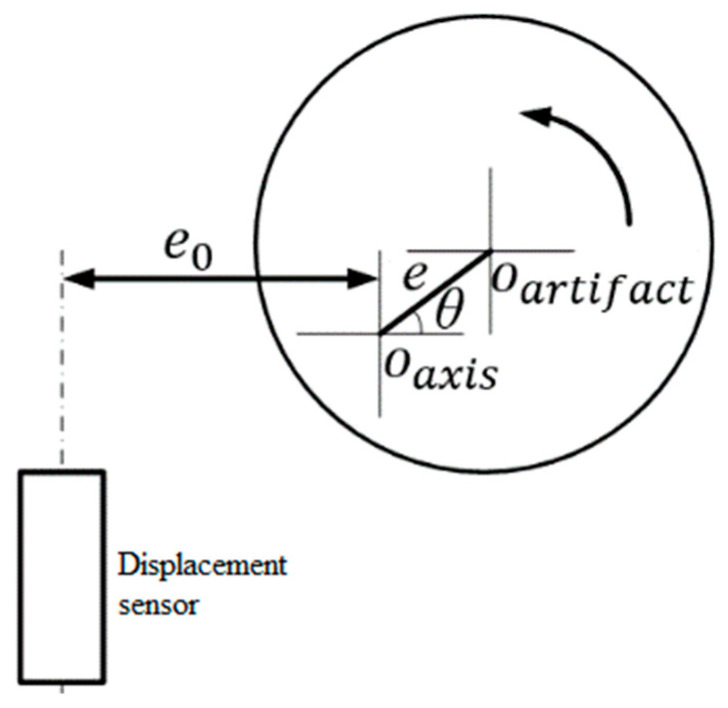
Tangential offset caused by eccentricity error.

**Figure 11 micromachines-13-02121-f011:**
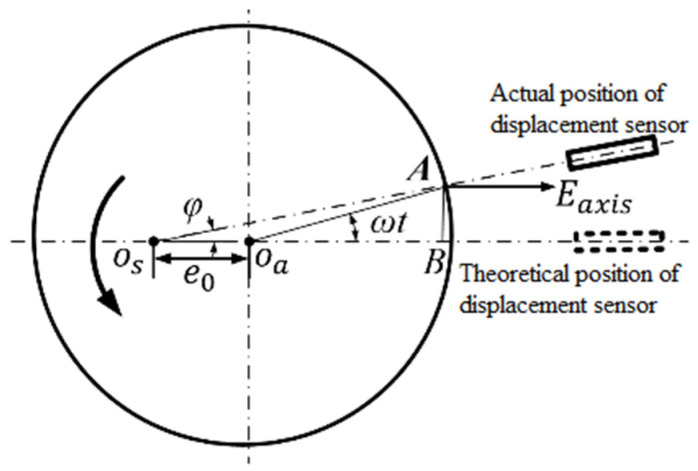
Effect of alignment error.

**Figure 12 micromachines-13-02121-f012:**
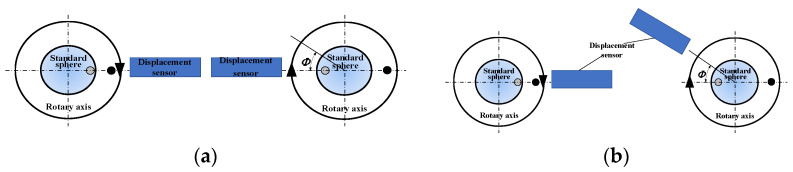
Positioning deviation (**a**) Standard sphere positioning deviation, (**b**) Probe positioning deviation.

**Figure 13 micromachines-13-02121-f013:**
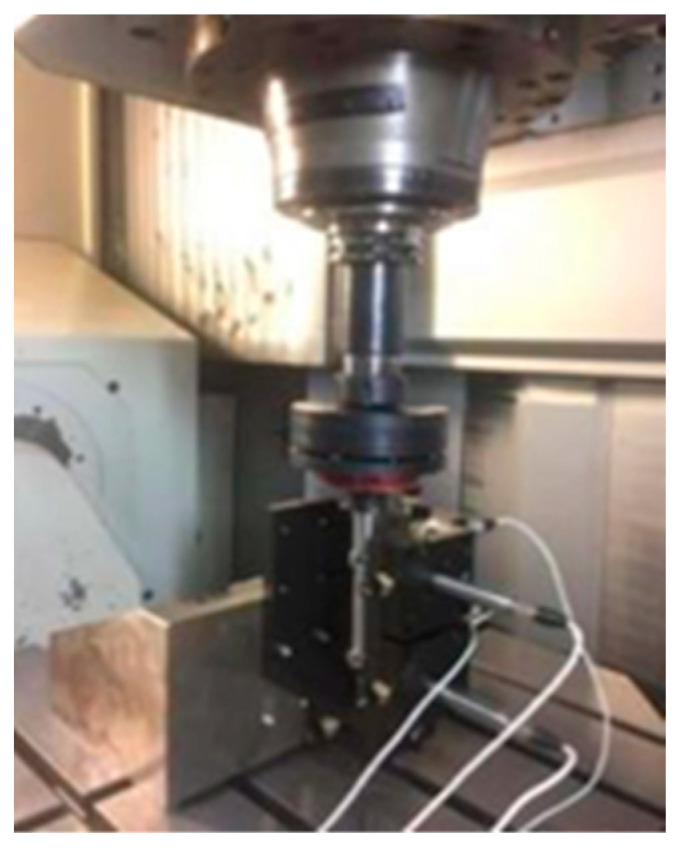
Developed the test on site.

**Figure 14 micromachines-13-02121-f014:**
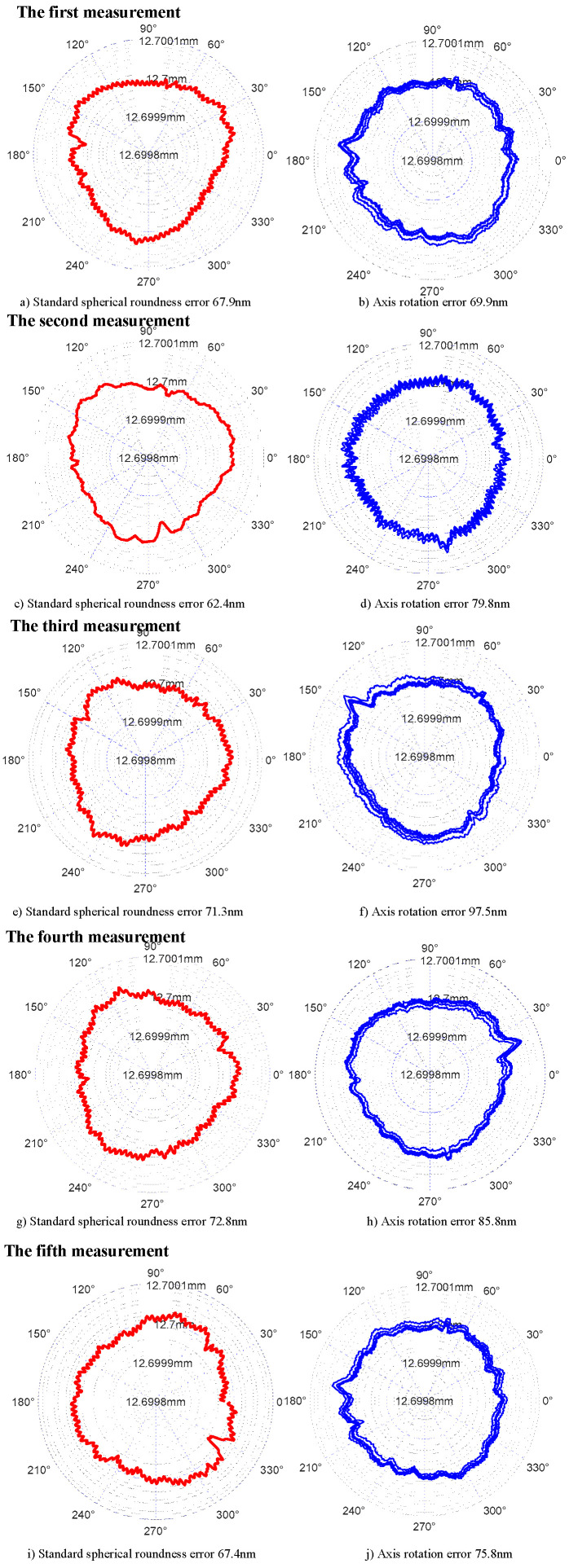
Repeatability test results of measurement system.

**Figure 15 micromachines-13-02121-f015:**
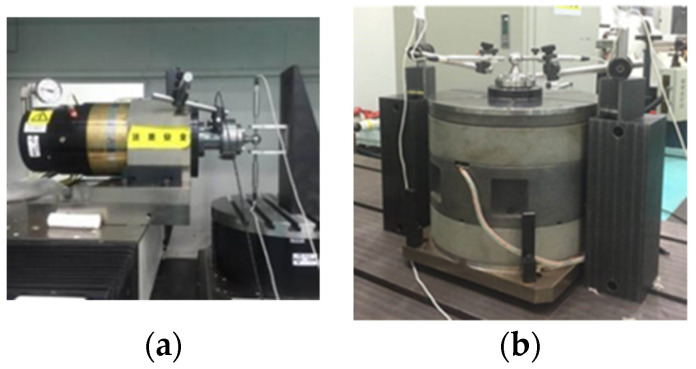
Test system and software developed (**a**) Test in the horizontal direction; (**b**) Test in the vertical direction.

**Figure 16 micromachines-13-02121-f016:**
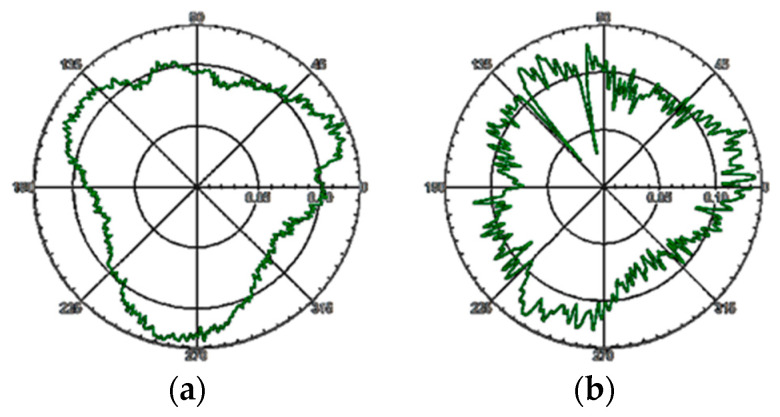
Comparison of different measurement objects (**a**) Roundness error for s1, (**b**) Roundness error for s2, (**c**) Roundness error for s3, (**d**) Roundness error for Turntable, (s1 is aerostatic spindle 1, s2 is aerostatic spindle 2, s3 is aerostatic spindle 3).

**Figure 17 micromachines-13-02121-f017:**
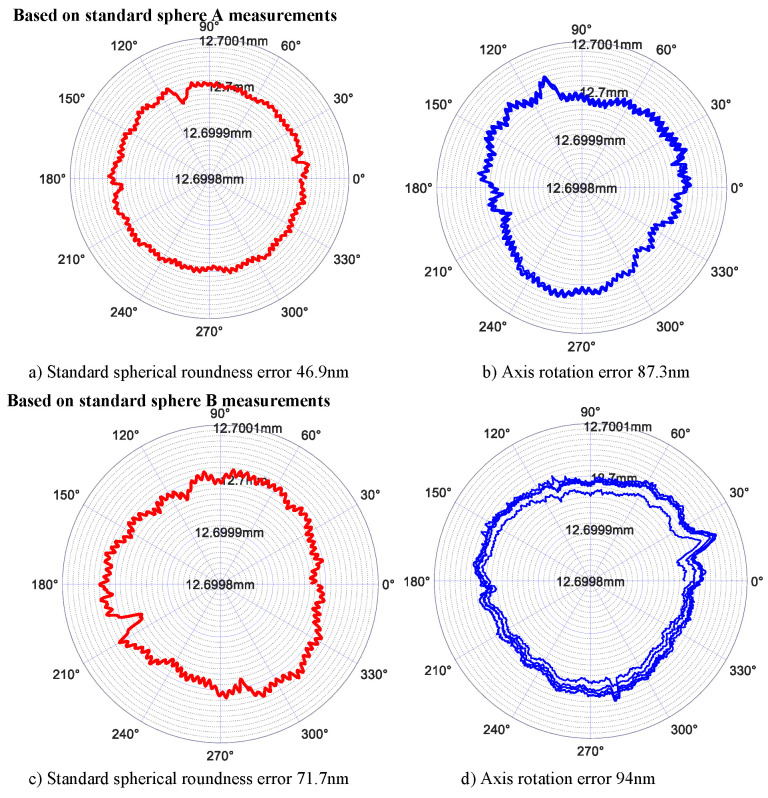
Comparison of different measurement objects.

**Table 1 micromachines-13-02121-t001:** Repeatability test results of measurement system (standard sphere A) (nm).

Number of Measurement	1	2	3	4	5
Standard spherical roundness error	67.9	62.4	71.3	72.8	67.4
Axis rotation error	69.9	79.8	97.5	85.8	75.8

**Table 2 micromachines-13-02121-t002:** Standard sphericity separation error (standard sphere b) (nm).

Measurement Object	Turntable	Main Shaft 1	Main Shaft 2	Main Shaft 3
Sphericity separation result	53 nm	60 nm	59 nm	66.4 nm
Sphericity calibration value	70 nm			

**Table 3 micromachines-13-02121-t003:** Standard sphericity separation error (standard sphere B) (nm).

Number of Measurements	1	2	3	4	5	Average Value
Using standard sphere A	112.6	108.0	94.4	87.3	98.8	100.2
Using standard sphere B	99.7	86.1	94	85.6	115.9	96.2

## Data Availability

Not applicable.

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
