# Peer review of "Radial Error Motion Measurement and Its Uncertainty Estimation of Ultra Precision Axes of Rotation with Nanometer Level Precision"

_micromachines, 2022, doi:10.3390/mi13122121_

Round 1
Reviewer 1 Report
In this paper, the measurement and evaluation of the radial error of a nanometer-scale ultra-precision shaft based on a capacitive displacement probe is investigated. It analyzes the influence of nonlinear capacitance displacement probe, probe misalignment, workpiece ball eccentricity and other factors on the error, proposes the complete process and main uncertainty components of the comprehensive evaluation of measurement uncertainty, and establishes the uncertainty evaluation mathematical model of the inversion method. However, there are some points that must be improved/clarified:
1. What are the axis names and units in Figure 9? Offset in which direction?
2. In Section 2.2.2, the standard deviation of the roundness error and axial motion error of the standard sphere is calculated. What is the detailed calculation process?
3. There are some grammatical errors in the paper, please correct them. Also, there are some formatting issues that need to be corrected. (1) There is an extra period at the end of sentence 1.2.2 on page 3. (2) There should be some gap between numbers and units. (3) The characters in Fig. 4a on page 4 are not legible.
Reviewer 2 Report
Dear Authors,
The paper presents a measurement technique for measuring the radial error motion and its uncertainty estimation of ultra-precision axis with nanometer level precision. In ultra-precision manufacturing, radial error and position displacements are important terms and directly influence the machining conditions as well as the quality of the machining outcome. The work could be useful in the field of ultra-precision machining as a technique for improving the measurement accuracy of radial errors.
Please find my comments below:
1. Please add a Nomenclature table
2. Please add a discussion section before the conclusions section while discussing/justifying the results of the performed measurements/estimations as well as the application/usefulness of the presented error detection system in improving the measurement accuracy of the radial errors.
3. Figures: It is not sufficient to include a figure without addressing it in the text. The figures should be addressed in the text under related subsections and information/details of the figures need to be added to the text.
4. Figures 5 and 6 need to be addressed in the text under the related paragraph.
5. Figure 16 need to be edited. Which one is a? which one is b, c, d?
6. “According to the above figure, when the same standard ball is….” Figure number should be mentioned.
7. Figure 17: Please use caption a b c d to better illustrate/present the results
8. Current format of the text is not readable. There are several errors in the text. Extensive editing of English language and style is required and all the issues need to be resolved. A few examples of the text errors are listed below:
- In some parts of the text “ultra-precision” term is used. In some other parts, “ultra precision” is used. Please replace the second format and only use “ultra-precision”.
- “The first-order error can be eliminated by eliminating the fundamental frequency, while the second-order error cannot be eliminated by mathematical methods. If the eccentricity error e=5μm. When the second-order error will reach 10nm. Therefore, it is necessary to control the eccentricity error of the standard ball by certain means to make it as small as possible.”
- “According to the analysis and test in Chapter 2.1 of this paper”. Section 2.1. is correct
- "This method can easily obtain the axis rotation error, but the standard ball, the standard ball, the standard mandrel and so on can be used to measure the rotation error of the spindle."
Round 2
Reviewer 1 Report
I suggest it can be published in this journal.
Reviewer 2 Report
Dear Authors,
Thank you for revising the paper and addressing my comments.
The paper could be suggested for acceptance.